# Anticholinergic Burden and Safety Outcomes in Older Patients with Chronic Hepatitis C: A Retrospective Cohort Study

**DOI:** 10.3390/ijerph17113776

**Published:** 2020-05-26

**Authors:** Patricia Amoros-Reboredo, Dolors Soy, Marta Hernandez-Hernandez, Sabela Lens, Conxita Mestres

**Affiliations:** 1Pharmacy Service Hospital Clínic de Barcelona, 08036 Barcelona, Spain; patri.amoros@gmail.com; 2Pharmacy Service Division of Medicines Hospital Clínic de Barcelona, University of Barcelona, IDIBAPS, 08036 Barcelona, Spain; DSOY@clinic.cat; 3School of Health Sciences Blanquerna, University Ramon Llull, 08025 Barcelona, Spain; martahh1@blanquerna.url.edu; 4Liver Unit Hospital Clínic de Barcelona, University of Barcelona, IDIBAPS, 08036 Barcelona, Spain; SLENS@clinic.cat; 5Centro de Investigación Biomédica Red de Enfermedades Hepáticas y Digestivas (CIBERehd), 28029 Madrid, Spain

**Keywords:** ageing, patient safety, drug use, adverse events, polypharmacy

## Abstract

*Aim*: Older patients with chronic hepatitis C infection starting direct-acting antivirals (DAAs) are frequently prescribed multiple medications that may be categorized as inappropriate. Anticholinergic burden has been shown to be a predictor of adverse health and functional outcomes. Different scales are available to calculate anticholinergic burden. The aim of this study was to determine the prevalence of anticholinergic medication among older patients treated with DAAs and the risk factors associated using the Anticholinergic Cognitive Burden (ACB) scale, the Anticholinergic Risk Scale (ARS) and the Anticholinergic Drug Scale (ADS) and analyze the resulting safety consequences. *Methods*: Observational, retrospective cohort study of consecutive patients ≥65 years old receiving DAAs and taking concomitant medication. This study was conducted in accordance with the Strengthening the Reporting of observational studies in Epidemiology Statement. *Results*: 236 patients were included. The average age was 71.7 years, 73.3% cirrhotic, and 47% patients took ≥5 medicines. According to the ACB, ARS and ADS scales, 35.2% (*n* = 83), 10.6% (*n* = 25) and 34.3% (*n* = 81) of the patients were treated with anticholinergic medication. Two hundred-and-six (86%) patients presented any adverse events (AEs) during therapy. ARS scale showed a significant relationship between presence of anticholinergic medication and AEs. A large number of patients suffered anticholinergic events, with more events per patient in patients taking anticholinergic drugs. *Conclusions*: Older hepatitis C chronic patients are exposed to potentially inappropriate polypharmacy and anticholinergic risk, according to the ACB, ARS and ADS scales. The three scales showed different results. Only the ARS scale was associated with AEs, but the rate of anticholinergic effects per patient was significantly higher in patients with anticholinergic drugs, regardless of the scale used. Consider quality of pharmacotherapy when starting DAA with a multidisciplinary approach could improve health outcomes.

## 1. Introduction

Elderly patients are particularly susceptible to adverse drug effects, mostly due to multiple comorbidities and cognitive and functional deficit, a high prevalence of multiple drugs, and age-related changes in pharmacokinetics and pharmacodynamics [1,2,3].

Drug prescription is generally based on evidence-based clinical guidelines for a single condition, and does not usually take into account multimorbidity. As a result, patients are often prescribed multiple medications by specialists guided by specific guidelines for their pathology, with limited consideration of comorbidities and concurrent medications, which in combination makes managing a disease difficult and likely to cause harm [4,5]. Commonly, the use of five or more medications is defined as polypharmacy; however, this definition is unable to distinguish between appropriate and inappropriate medication use [4]. Polypharmacy in older adults is linked to inappropriate drug treatment and has been associated with adverse drug events, hospitalization, and even death, for this reason, the World Health Organization (WHO) has described polypharmacy as a significant public health challenge [5,6]. In this way, it is essential to identify patients at risk to reduce the impact of these events both in terms of cost and quality of care [7]. Several tools have been defined in the literature to identify patients at risk, including anticholinergic burden scales [4,8].

Acetylcholine is a chemotransmitter that is involved in many physiological processes [9]. A large number of drugs have anticholinergic activity that is unrelated to their primary mode of action, acting on muscarinic receptors located in the brain, heart, exocrine glands and smooth muscle, which are described as anticholinergic or antimuscarinic adverse effects [9,10,11]. These effects include dry mouth and eyes, blurred vision, dry skin, constipation, tachycardia, urinary retention, sedation, restlessness, anxiety, confusion and delirium [9,11,12]. The prevalence of anticholinergic use in older patients ranges from 8% to 37% [13]. The accumulation of higher levels of exposure due to one or more anticholinergic medications, and the attendant increased risk of medication-related adverse effects, is termed anticholinergic burden [10]. Older patients are more susceptible to these effects due to decreased renal and hepatic metabolism, as well as the increased permeability of the blood–brain barrier.

Recently, the efficacy and safety of direct acting antivirals (DAA) in elderly patients has been a topic of interest [14,15,16,17,18,19,20,21,22,23,24,25,26,27]. These are highly effective also in older patients with hepatitis C-related liver disease, comorbidities and concomitant medications causing drug–drug interactions (DDIs) [14,15,20,21,22]. However, in a recent study from our group, a larger number of patients (86%) suffered adverse events (AEs) and the rate of serious AEs (SAEs) increased with the number of concomitant medications and the severity of comorbidity, highlighting that older patients with high comorbidity and polypharmacy were more prone to suffer SAEs [22]. Other studies also concluded that incidence of adverse events is higher in patients with comorbidities [20,25]. Thus, clinicians treating patients with DAA may be unaware that many medications frequently used to treat common chronic conditions might have weak anticholinergic properties. These drugs, when used singly or in combination, may result in adverse effects through the accumulation of anticholinergic burden and may increase the risk of suffering AEs [28].

Multiple scales exist for the measurement of the anticholinergic burden but they show differences in terms of the medications they include and the anticholinergic potency score for each medication [10,12,29]. Among the most used are the Anticholinergic Cognitive Burden (ACB) scale, the Anticholinergic Risk Scale (ARS) and the Anticholinergic Drug Scale (ADS) [12,30,31,32,33,34].

The aim of this study was to determine the prevalence of anticholinergic medication among older patients treated with DAAs and the risk factors associated, using the ACB, ARS and ADS scales, and analyze the resulting safety consequences.

## 2. Methods

This was an observational, retrospective cohort study conducted in a third-level university hospital (Hospital Clínic, Barcelona, Spain).

Over 11 months, all patients aged ≥ 65 years starting antiviral therapy in routine clinical practice and taking at least one concomitant medication were included in the analysis. The study was approved by the local Clinical Research Ethics Committee (HCB/2016/0872). All data were anonymized. This study was conducted in accordance with the STROBE (Strengthening the Reporting of observational studies in Epidemiology) Statement.

### 2.1. Variables

Demographic and clinical variables were collected, using data from the electronic records: age, gender, baseline comorbidities and virological and clinical factors related to HCV infection (genotype, degree of fibrosis, liver function, liver transplant, esophageal varices, antiviral therapy and treatment duration). Cirrhosis was diagnosed based on a liver stiffness value > 12.5 kPa or the presence of portal hypertension (esophageal varices, clinical decompensation or a hepatic venous pressure gradient ≥ 6 mmHg) [35]. Electronic records contained rich clinical records with high-quality data, including diagnosis codes, pharmacy, lab test results hospital admissions, emergency and ambulatory visits.

### 2.2. Comorbidities

Comorbidities were analyzed using the Clinical Risk Group (CRG) [36]. The CRG are a population classification system that uses inpatient and ambulatory diagnosis and procedure codes, pharmaceutical data and functional health status to assign each individual to a single, severity-adjusted group. These groups report the individual health status and illness severity level, contemplating whether the disease is acute or chronic and the number of organs or systems affected [22,36,37]. Those presenting a high comorbidity index, which corresponded to chronic diseases in 2 or more affected organs and a high severity level (≥6/05), were selected for closer analysis.

### 2.3. Exposure to Polypharmacy and Anticholinergic Medication

Pharmacologic variables included: Number of concomitant medication at starting DAAs (patients were classified as “oligopharmacy” if <5 drugs, “moderate polypharmacy” if 5–9 drugs and “excessive polypharmacy” if ≥10 drugs) [30]. Qualitative classification of drugs under the Anatomical Therapeutic Chemical (ATC) classification System. All drug–drug interactions (DDIs) were assessed according to the University of Liverpool website: http://www.hep-druginteractions.org/, with a multidisciplinary approach managed by hepatologists, nurses and clinical pharmacists.

Anticholinergic burden was assessed according to the “Web Portal Software Anticholinergic Burden Calculator”: http://www.anticholinergicscales.es/. These scales rank anticholinergic agents into 3 categories: category 1: agents with a low effect; category 2: those with a moderate effect; category 3: those with a high effect [32,33,34].

### 2.4. Assessing Clinical Outcomes

Safety assessments included laboratory data, physical examinations, evaluation of vital signs and the reporting of adverse events (AEs) specified at the computerized clinical history during DAAs treatment were collected. AEs were defined, according to Good Clinical Practice, as any untoward medical occurrence which does not necessarily have a causal relationship with the treatment [38]. Patients had been followed up until 12 weeks after the end of treatment with clinical and laboratory evaluations, by the same team of physician, nurse and pharmacist. Safety outcomes were analyzed according to the presence or not of anticholinergic medications, cirrhosis, high comorbidity, and the use of ribavirin and DDIs between DAA and concomitant medications.

Any event related to anticholinergic effects were classified as anticholinergic (Ach) event, specifically, events related to dry mouth and eyes, blurred vision, dry skin, constipation, tachycardia, urinary retention, sedation, restlessness, anxiety, confusion and delirium.

### 2.5. Statistical Analysis

Quantitative variables were compared by Student’s t-test. Results were analyzed using the χ^2^-test for comparison between qualitative variables. In addition, different groups of patients according to anticholinergic burden were considered: 1–2 points (low anticholinergic burden) and ≥3 points (high anticholinergic burden) according to ADS, ACB and ARS scales. If data of anticholinergic risk was missing for one drug it was considered as no risk. If no anticholinergic risk data were available for a drug, it was considered non-risk (0 points). *p* < 0.05 was considered statistically significant.

## 3. Results

### 3.1. Baseline Characteristics

A total of 236 patients aged ≥65 years treated with DAAs combination regimens were included. The mean age was 71.7 years (SD 4.7 years) and 62.3% were female (*n* = 147). A summary of baseline characteristics is provided in Table 1. Genotype 1b was predominant (*n* = 210; 89%), almost half of patients (*n* = 117; 49.6%) had failed to a prior interferon-based antiviral treatment, 22 patients (9.3%) had undergone previous liver transplantation and 173 had advanced liver disease or cirrhosis (73.3%). High comorbidity index (CRG ≥ 06/5) was present in 105 patients (44.5%). Arterial hypertension and diabetes mellitus were the most frequent comorbidities (59.3% and 22.5%, respectively). Forty-seven patients (19.9%) had previous cancer history, all in complete remission, 18 of them hepatocellular carcinoma. All patients were treated with interferon-free regimens; the most common were ombitasvir plus paritaprevir plus ritonavir (3D) or sofosbuvir (SOF)/ledipasvir (LDV). The DAA treatment schedule was: 3D ± ribavirin (RBV) 118 patients (50.0%), SOF/LDV ± RBV 100 patients (42.4%), SOF + RBV 10 patients (4.2%), SOF/simeprevir (SMV) ± RBV 5 patients (2.1%) and SOF/daclatasvir (DCV) ± RBV 3 patients (1.3%). RBV was taken by 151 patients (64.0%). The treatment duration scheduled was 12 weeks in most patients (193 patients, 81.8%).

In addition to the HCV treatment, patients took an average number of 5 concomitant medication (SD 2.8 drugs), with 47% patients with moderate or excessive polypharmacy. The most used drugs were diuretics (53%), psycholeptics (37%), drugs for acid related disorders (36%) and agents acting on the rennin-angiotensin system (36%). Potential DDIs were present in 156 patients (66%).

### 3.2. Anticholinergic Burden and Associated Risk Factors

According to the ACB, ARS and ADS scales, 35.2% (*n* = 83), 10.6% (*n* = 25) and 34.3% (*n* = 81) of the patients were being treated with anticholinergic medication (average drugs per patient ± SD: 0.5 ± 0.7; 0.1 ± 0.4; 0.5 ± 0.8), with 8.9% (*n* = 21), 4.2% (*n* = 10) and 8.9% (*n* = 21) of patients having a high anticholinergic burden, respectively (Table 2, Table 3 and Table 4).

Patients under anticholinergic therapy had a higher comorbidity index (ACB: *p* = 0.01, ARS: *p* < 0.01, ADS: *p* = 0.1) (Table 2, Table 3 and Table 4, respectively).

No differences were observed among patients taking and not taking anticholinergic medication for the remaining demographic or clinical variables, including potential DDIs. Similarly, no differences were observed when comparing the groups with a high anticholinergic burden with those with a lower anticholinergic burden for any variable.

Drugs classified as nervous system drugs (ATC = *N*) were the most commonly considered inappropriate by the three scales and taken by the majority of patients. Figure 1 presents the proportion of patients with drug exposure for the principal pharmacologic classes of medications for each score.

### 3.3. Safety Outcomes

AEs were reported in 206 patients (87.3%). The most reported AEs were fatigue (97 patients, 41.1%), gastrointestinal symptoms (63 patients, 26.7%), skin complaints (53 patients, 22.5%) and anemia (48 patients, 20.3%). Nine patients required hospital admission during study period, and two of them died. Three patients discontinued treatment. Two patients died out-of-hospital and the cause of death was unrelated to the DAA therapy according to physician criteria. Only one patient discontinued treatment due to adverse events, after a 3D overdose that caused him malaise, edema and gastrointestinal complaints.

Only the ARS scale showed a significant relationship between the presence of anticholinergic medication and AEs, and high anticholinergic burden and number of AEs per patient (Table 2, Table 3 and Table 4). However, by using this scale, no differences were observed between taking and not taking anticholinergic medication, or when comparing the groups with a high anticholinergic burden with those with a lower anticholinergic burden, for hospital admission or death.

Anticholinergic events were reported in 99 patients (41.9%), and were related to dry skin (53 patients, 22.5%), confusion (25 patients, 10.6%), restlessness (20 patients, 8.5%), tachycardia (13 patients, 5.5%), urinary retention (12 patients, 5.1%), dry mouth (8 patients, 3.4%), anxiety (4 patients, 1.7%) and constipation (1 patient, 0.4%).

Regarding anticholinergic events, no differences were observed between taking and not taking anticholinergic medication, but patients with anticholinergic burden suffered more anticholinergic events, regardless of the scale used. The ADS scale also showed a significant relationship between high anticholinergic burden and rate of anticholinergic events per patient (Table 2, Table 3 and Table 4).

Although AEs were more frequently reported in cirrhotic patients compared to non-cirrhotic, the difference did not reach statistical significance (89% vs. 82.5%, *p* = 0.19). Hospital admission was also higher in cirrhotic patients (1.6% in non-cirrhotic patients vs. 4.6% in cirrhotic, *p* = 0.29), and all patients who died where cirrhotic (0% non-cirrhotic vs. 2.3% of cirrhotic patients died, *p* = 0.23). Predicted clinically significant DDIs were present in 156 patients (66.1%). Potential DDIs between DAAs and comedications were no related with safety poorer outcomes. Of 80 patients without potential DDIs, 70 suffered AEs (87.5%), 5 were admitted to hospital (6.3%), and 1 died (1.3%). Of patients with potential DDIs, 136 suffered AEs (87.2%), 4 were admitted to hospital (2.6%), and 3 died (1.9%) (AEs *p* = 0.95, hospital admission *p* = 0.16 and death *p* = 0.74).

A total of 151 were treated with ribavirin, and suffered more AEs compared to patients without ribavirin, but the difference did not reach statistical significance (89.4% vs. 83.5%, *p* = 0.19). Dose reduction or discontinuation of the RBV daily dose was observed in 71 of them (47%).

## 4. Discussion

This real world population study shows the prevalence of anticholinergic medicines in older chronic hepatitis C patients with advanced liver disease and high comorbidity starting DAAs, by using three validated scales—the ADS, ACB and ARS [32,33,34] and analyses the results of these scales on adverse outcomes.

We found a high prevalence of patients taking anticholinergic drugs using ACB and ADS scales (35.2% and 34.3%). We also found a relationship between comorbidity and drugs with a greater anticholinergic effect. Patients with anticholinergic drugs had more comorbidity (CRG ≥ 6/05) [39]. The presence of moderate and excessive polypharmacy (≥5 and ≥10 drugs) has been found to be a risk factor for the presence of anticholinergic medication; data in concordance with those observed by Sevilla et al. [30]. Only the ARS scale showed significant relationship between anticholinergic burden and AEs, also in the rate of AEs per patient. The rate of anticholinergic events per patient was significantly higher in patients using anticholinergic drugs according to the three scales. Prevalence of patients taking anticholinergic drugs was statistically significantly higher (*p* < 0.0001) on the ACB and ADS scales (35.2% and 34.3%) than on the ARS scale (10.6%). Anticholinergic medications are considered inadequate for geriatric patients [39]. Mechanisms potentially explaining the association between anticholinergic medicines and adverse events include age-related changes in pharmacokinetics and pharmacodynamics, as well as increased permeability of blood–brain barriers [40]. Additionally, in chronic hepatitis C patients, advanced liver impairment might predispose to an increased risk of adverse effects from anticholinergic drugs.

Our data show similar results using ACB and ADS scales but different with ARS, in accordance with previous studies [30,31,41]. According to the ACB and ADS scales, 8.9% of patients had a high anticholinergic risk, statistically significantly higher than the 4.2% obtained using the ARS scale (*p* = 0.04). ACB scale includes high-ceiling diuretics (such as furosemide) in the group of anticholinergics, and our population has advanced hepatic disease, with extensive use of this kind of diuretics, resulting in a high use of anticholinergic medication. In addition, both ACB and ADS scale include paroxetine and quetiapine as moderate or high anticholinergic burden. Both drugs were taken by a large number of our study patients mainly because antipsychotics are widely used in geriatrics for different disorders and quetiapine and paroxetine, specifically, had been shown to be efficacious in generalized anxiety disorder, which is one of the most common psychiatric disorders in primary care [42,43]. In addition, estimates from the literature suggest that psychiatric disorders were common comorbidities among HCV-infected patients [44]. Moreover, anxiolytics and antidepressants have been associated with adverse outcomes in the elderly, such as falling and hospital admissions [45].

ARS is one of the tools associated with the highest number of patient-related outcomes, being associated with hospitalization, mortality, falls and functional decline [4]. By using the ARS scale, we found a relationship between the presence of anticholinergic drugs and the development of AEs in patients receiving DAAs. Although no significant differences were found, there also was a higher presence of hospital admission in patients with higher anticholinergic burden, according to ACB and ARS scales. These results are in accordance with Wan-Hsuan et al., who found an association between the ARS scale and all-cause admission to hospital. They also consider polypharmacy and anticholinergic burden as quality indicators of polypharmacy in older adults [46]. Hsu et al. also showed good response relationships between anticholinergic burden using the ACB scale and a variety of adverse outcomes in older adults. This study shows that anticholinergic scales tend to be an indicator of prescriptions with a risk for AEs, also described by Jean-Bart et al [45]. The prolonged and cumulative administration of these drugs makes them especially vulnerable to anticholinergic adverse effects because of the advanced age and frailty [47]. Anticholinergic burden has also shown to be a good predictor of adverse health and functional outcomes [48,49,50,51,52]. Hence, anticholinergics are generally categorized as potentially inappropriate medications for use in older adults and the estimation of the anticholinergic drug burden has been suggested as a way of reducing the risk of secondary cognitive decline of drug therapy and of optimizing polypharmacy in the elderly [53,54]. In our study, mortality was slightly higher in patients taking medication with anticholinergic burden according to ADS scale. A recent meta-analysis indicated an association between anticholinergic exposure and higher risk of mortality using the ACB, ARS and ADS scales [12]. Recently, Lozano-Ortega et al. described that the ACB and ADS scales were well suited for implementation in observational studies where anticholinergic exposure needs to be quantified [10].

Antiviral treatment was generally well tolerated, with only one patient discontinuing treatment due to adverse events. Nevertheless, two patients died during treatment even though the cause of death was unrelated to the DAA therapy. Fatigue was the top reported AE, in concordance with the data observed by Huang et al. [55]. In their study, DAA were found to have higher reporting rates in a few AEs—e.g., fatigue and abdominal pain—using data from the U.S. Food and Drug Administration Adverse Event Reporting System, and diarrhea, abdominal pain, pain and pyrexia using data of Electronic Health Records [55]. In our study population, the prevalence of skin complaints and gastrointestinal effects was higher than that found by Villani et al. in a recent meta-analysis [24]. Anemia was frequent in our population, which is typically associated with RBV use, similar to other studies [14,24,56]. In 47% of patients, RBV dose reduction or discontinuation was needed, which is a higher proportion of cases than reported by Conti et al. [15].

Although the rate of DDIs in our study was quite high (66%), there were no clinically significant interactions related to AEs, hospital admission nor death. It was in part because a meticulous DDIs assessment before treatment initiation and careful monitoring were realized by the multidisciplinary team to avoid DDIs.

Possible anticholinergic effects were observed in a large number of patients, in almost half of the patients who suffered any adverse event. Effects probably related to dry skin, confusion and restlessness were the most observed. It should be noted that effects related to dry skin may also be as a result of the antiviral treatment described above as skin complaints. Although no relationship was seen between anticholinergic effects and anticholinergic burden according to any of the scales, a significant relationship was observed between the number of anticholinergic effects and the anticholinergic burden with all scales.

The main strengths of our study are the analysis of polypharmacy, its appropriateness using anticholinergic scales, and safety-related outcomes in a real-world population of older patients receiving DAA therapy including a large number of cirrhotic patients (*n* = 236 patients) with high comorbidity, polypharmacy and, therefore, risk of DDIs. To our knowledge, this is the first study assessing the appropriateness of concomitant therapy and its clinical safety outcomes in older chronic hepatitis C patients receiving DAAs. No studies are available on HCV older patients receiving DAAs to compare the results on the use of anticholinergic agents. Variability between scales in the proportion of patients identified as taking anticholinergic drugs and the grade of anticholinergic burden was found. As other authors have suggested, the use of multiple anticholinergic burden measures on the same population reduces risk from heterogenicity and increases confidence in making comparisons [12,41]. Nevertheless, the study has some limitations. Firstly, its retrospective nature may imply some selection bias. Since it includes out-hospital patients, even though all medicines dispensed during the study period were included in the analyses, it could not be ascertained whether the dispensed medicines were actually consumed. Secondly, our findings only indicate association, and were susceptible to residual confounding. Casual relation between anticholinergic burden and AEs was not proven. Finally, electronic clinical records data are collected based on routine medical practice, not for pharmacovigilance research, and some AEs, such as fatigue, can be caused by HCV infection itself, or other long-term AEs may not be observed due to the relatively short follow-up time.

In concordance with Park et al., our study shows differences in the total prevalence of anticholinergic use evaluated by the different scales, and there is no standardized rating scale for the measurement of anticholinergic burden; therefore, further research is necessary to develop a useful and comprehensive tool identifying medications with anticholinergic properties [54]. However, as suggested by Hanlon et al., anticholinergic risk scales are easy and useful for identifying patients at risk of adverse effects, regardless of the scale used [41]. Sessa et al., found a proportion of preventable adverse drug reactions involving DAAs, suggesting that it would be a target for improvement [56]. Anticholinergic medications are a potentially modifiable risk factor for the prevention of adverse events and one may hypothesize that recognizing the use of anticholinergic drugs—and therefore potentially inappropriate polypharmacy—by means of these scales, could help in identifying older patients with comorbidities at risk of adverse events when starting antiviral therapy [57]. As suggested by Merle et al., limiting drug prescription to essential medications and periodically re-evaluating all use of drugs in the elderly could reduce the prevalence of AEs [58]. The study highlights the need to revise concomitant conditions and the treatment of hepatitis C chronic patients, and therapy initiation presents a window of opportunity where a multidisciplinary could make patient-centered decisions. Anticholinergic burden tools—probably ARS as the most clinically relevant—might be recommended as a complementary procedure to comprehensive geriatric assessment, with a multidisciplinary approach in patients starting any treatment, mainly chronic hepatitis C patients with advanced liver disease, high comorbidity, polypharmacy and risk of DDIs, when starting DAAs [4].

In fact, HCV therapy initiation presents a window of opportunity for overall treatment review—particularly for those prescribed multiple medicines, or taking combinations of medicines with a higher risk of adverse effects, including enhanced coordination of care between hepatologists, clinical pharmacists and other subspecialists.

In conclusion, older hepatitis C chronic patients commonly had multiple comorbidities and used co-medications with potential anticholinergic effects, therefore they are exposed to inappropriate polypharmacy. The presence of anticholinergic drugs was associated with AEs using ARS scale. The rate of anticholinergic effects per patient was significantly higher in patients with anticholinergic drugs, regardless of the scale used. Clinicians treating older adults starting DAA should be aware of the risk associated with comorbidity and comedications that may increase the risk of AEs. To provide optimal antiviral treatment, a coordination of care between hepatologists and clinical pharmacists supported by a multidisciplinary team is needed.

## Figures and Tables

**Figure 1 ijerph-17-03776-f001:**
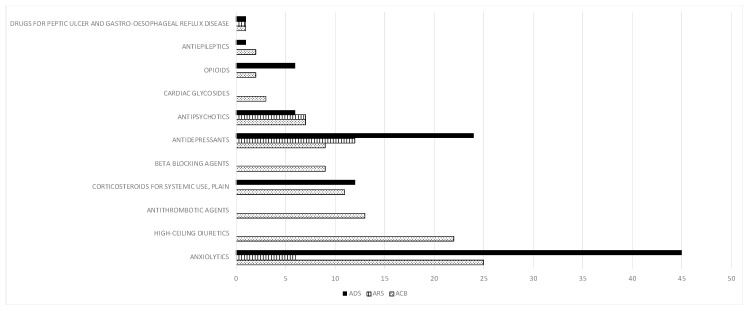
Medications prescribed to study patients with anticholinergic properties according to Anticholinergic Drug Scale (ADS), Anticholinergic Risk Scale (ARS) and Cognitive Burden Scale (ACB). *X*-axis: number of patients taking these drugs. ADS is shown solid, ARS lined and ACB dotted.

**Table 1 ijerph-17-03776-t001:** Baseline characteristics.

Variable	*N* = 236
Age, years	71.7 (4.7)
Female gender	147 (62.3%)
Fibrosi stage:	
f0–f1	6 (2.5%)
f2	16 (6.8%)
f3	41 (17.4%)
Cirrhosis	173 (73.3%)
Previous liver transplantation	22 (9.3%)
Clinical Risk Group (>06/5)	105 (44.5%)
Arterial hypertension	140 (59.3%)
Diabetes	53 (22.5%)
Cardiopathy	37 (15.7%)
Arrhythmia	15 (6.4%)
COPD	12 (5.1%)
Depression	27 (11.4%)
Tumor	47 (19.9%)
History of previous HCC	18 (7.6%)
Treatment experienced	117 (49.6%)
HCV genotype:	
1a	13 (5.5%)
1b	210 (89.0%)
2	9 (3.8%)
3	3 (1.3%)
4	1 (0.4%)
Oligopharmacy	124 (52.5%)
Moderate polypharmacy	101 (42.8%)
Excessive polypharmacy	11 (4.7%)
Patients with potential DDIs	156 (66.1%)

Data are given as average (SD) or as number cases (%). Hepatitis C virus (HCV), hepatocellular carcinoma (HCC), chronic obstructive pulmonary disease (COPD), drug–drug interactions (DDIs).

**Table 2 ijerph-17-03776-t002:** Anticholinergic risk factors and safety outcomes according to Cognitive Burden Scale (ACB).

Characteristics	Without Ach-Drugs	With Ach-Drugs	Comparison between Groups (*p*-Value)	Low ACB (1–2 Points)	High ACB (≥3 Points)	Comparison between Groups (*p*-Value) 2
No.patients	153	83		62	21	
Age, years *	72.2 (4.8)	70.7 (4.4)	**0.0191**	70.5 (4.1)	71.4 (5.1)	0.4169
Sex (female)	91 (59.5%)	56 (67.5%)	0.2269	42 (67.7%)	14 (66.7%)	0.9330
CRG ≥ 6/05 *	59 (38.6%)	46 (55.4%)	**0.0133**	32 (51.6%)	14 (66.7%)	0.2317
Mean drugs/patient *	4.0 (2.5)	5.7 (3.0)	**<0.0001**	5.5 (3.1)	6.3 (2.6)	0.2915
Oligopharmacy	93 (60.8%)	31 (37.3%)	**0.0006**	26 (41.9%)	5 (23.8%)	0.1407
Moderate polypharmacy	58 (37.9%)	43 (51.8%)	**0.0397**	29 (46.8%)	14 (66.7%)	0.1169
Excessive polypharmacy	2 (1.3%)	9 (10.8%)	**0.0009**	7 (11.3%)	2 (9.5%)	0.8197
Potential DDIs	95 (62.1%)	61 (73.5%)	0.0883	46 (74.2%)	15 (71.4%)	0.8028
AEs (*N* = 206)	133 (86.9%)	73 (88%)	0.8090	52 (83.9%)	21 (100%)	0.0513
AEs per patient *	2.9 (1.8)	2.9 (1.9)	1.0000	2.8 (1.7)	3.3 (2.3)	0.3093
Hospital admission (*N* = 9)	5 (3.3%)	4 (4.8%)	0.5671	2 (3.2%)	2 (9.5%)	0.2457
Death (*N* = 4)	3 (2.0%)	1 (1.2%)	0.6523	1 (1.6%)	0 (0%)	0.5622
Ach events (*N* = 99)	63 (41.2%)	36 (43.4%)	0.7442	25 (40.3%)	11 (52.4%)	0.3365
Ach events per patient *	1.3 (0.6)	1.5 (0.8)	**0.0312**	1.4 (0.8)	1.8 (1.0)	0.0671

Data are given as average (SD) or as number cases (%); * mean (SD); Cognitive Burden Scale (ACB), anticholinergic drugs (Ach-drugs), clinical risk group (CRG), drug–drug interactions (DDIs), adverse events (AEs). Bold values denote statistical significance at the *p* < 0.05 level.

**Table 3 ijerph-17-03776-t003:** Anticholinergic risk factors and safety outcomes according to Anticholinergic Risk Scale (ARS).

Characteristics	Without Ach-Drugs	With Ach-Drugs	Comparison between Groups (*p*-Value)	Low ARS (1–2 Points)	High ARS (≥3 Points)	Comparison between Groups (*p*-Value) 2
No.patients	211	25		15	10	
Age, years *	71.8 (4.7)	70.4 (4.8)	0.1613	69.8 (3.5)	71.4 (6.3)	0.4221
Sex (female)	130 (61.6%)	17 (68.0%)	0.5333	9 (60.0%)	8 (80.0%)	0.3035
CRG ≥ 6/05	87 (41.2%)	18 (72.0%)	**0.0035**	11 (73.3%)	7 (70.0%)	0.8600
Mean drugs/patient *	4.4 (2.8)	6.1 (2.5)	**0.0041**	5.7 (2.6)	6.6 (2.3)	0.3845
Oligopharmacy	118 (55.9%)	6 (24.0%)	**0.0026**	5 (33.3%)	1 (10.0%)	0.1903
Moderate polypharmacy	84 (39.8%)	17 (68.0%)	**0.0072**	9 (60.0%)	8 (80.0%)	0.3035
Excessive polypharmacy	9 (4.3%)	2 (8.0%)	0.4091	1 (6.7%)	1 (10.0%)	0.7706
Potential DDIs	138 (65.4%)	18 (72.0%)	0.5107	11 (73.3%)	7 (70.0%)	0.8600
AEs (*N* = 206)	181 (85.8%)	25 (100%)	**0.0442**	15 (100%)	10 (100%)	-
AEs per patient *	2.8 (1.8)	3.3 (2.2)	0.2071	2.5 (1.4)	4.4 (2.9)	**0.0383**
Hospital admission (*N* = 9)	7 (3.3%)	2 (8.0%)	0.2460	0 (0%)	2 (20.0%)	0.0768
Death (*N* = 4)	4 (1.9%)	0 (0%)	0.4879	0 (0%)	0 (0%)	-
Ach events (*N* = 99)	85 (40.3%)	14 (56.0%)	0.1334	6 (40.0%)	8 (80.0%)	0.0531
Ach events per patient *	1.3 (0.7)	1.8 (0.9)	**0.0012**	1.7 (0.5)	1.9 (1.1)	0.5418

Data are given as average (SD) or as number cases (%); * mean (SD); Anticholinergic Risk Scale (ARS), anticholinergic drugs (Ach-drugs), clinical risk group (CRG), drug–drug interactions (DDIs), adverse events (AEs). Bold values denote statistical significance at the *p* < 0.05 level.

**Table 4 ijerph-17-03776-t004:** Anticholinergic risk factors and safety outcomes according to Anticholinergic Drug Scale (ADS).

Characteristics	Without Ach-Drugs	With Ach-Drugs	Comparison between Groups (*p*-Value)	Low ADS (1–2 Points)	High ADS (≥3 Points)	Comparison between Groups (*p*-Value) 2
No.patients	155	81		60	21	
Age, years *	71.9 (4.8)	71.3 (4.6)	0.3561	71.6 (4.4)	70.6 (5.0)	0.3896
Sex (female)	92 (59.4%)	55 (67.9%)	0.2017	42 (70.0%)	13 (61.9%)	0.4965
CRG ≥ 6/05	63 (40.6%)	42 (51.9%)	0.0979	28 (46.7%)	14 (66.7%)	0.1167
Mean drugs/patient *	4 (2.7)	5.8 (2.7)	**<0.0001**	5.7 (2.8)	6.1 (2.4)	0.5613
Oligopharmacy	97 (62.6%)	27 (33.3%)	**<0.0001**	23 (38.3%)	4 (19.0%)	0.1084
Moderate polypharmacy	54 (34.8%)	47 (58.0%)	**0.0006**	31 (51.7%)	16 (76.2%)	0.0517
Excessive polypharmacy	4 (2.6%)	7 (8.6%)	**0.0383**	6 (10.0%)	1 (4.8%)	0.4684
Potential DDIs	99 (63.9%)	57 (70.4%)	0.3175	44 (73.3%)	13 (61.9%)	0.3279
AEs (*N* = 206)	136 (87.7%)	70 (86.4%)	0.7766	51 (85%)	19 (90.5%)	0.5291
AEs per patient *	2.9 (1.7)	2.9 (2.1)	1.0000	2.6 (1.9)	3.5 (2.6)	0.1168
Hospital admission (*N* = 9)	6 (3.9%)	3 (3.7%)	0.9396	1 (1.7%)	2 (9.5%)	0.1063
Death (*N* = 4)	2 (1.3%)	2 (2.5%)	0.5007	2 (3.3%)	0 (0%)	0.4022
Ach events (*N* = 99)	63 (40.6%)	36 (44.4%)	0.5751	25 (41.7%)	11 (52.4%)	0.3987
Ach events per patient *	1.3 (0.6)	1.6 (0.9)	**0.0025**	1.4 (0.9)	1.9 (0.9)	**0.0314**

Data are given as average (SD) or as number cases (%). * mean (SD); Anticholinergic Drug Scale (ADS), Anticholinergic drugs (Ach-drugs), Clinical Risk Group (CRG), Drug–drug interactions (DDIs), Adverse Events (AEs). Bold values denote statistical significance at the *p* < 0.05 level.

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
