# Peer review of "Anticholinergic Burden and Safety Outcomes in Older Patients with Chronic Hepatitis C: A Retrospective Cohort Study"

_ijerph, 2020, doi:10.3390/ijerph17113776_

Round 1
Reviewer 1 Report
The authors investigated the prevalence of anticholinergic medication among elder patients HCV infection treated with DAAs and the risk factors associated using three scales for the measurement of the anticholinergic burden and analyze the resulting safety consequences. ARS scale was exclusively associated with AEs. They concluded that older patients with HCV infection commonly had multiple comorbidities and used co-medications with potential anticholinergic effects. The purpose and result of the study is interesting, and it is well written, but the discussion is not impressive.
Author Response
Report 1
The discussion is not impressive
We sincerely appreciate all valuable comments and suggestions. The discussion section has been fully revised according to the reviewer's comments.
Reviewer 2 Report
The authors performed a study to investigate the prevalence of anticholinergic medication in older patients treated with Direct-acting antivirals.
Most HCV patients are 65 years and older and a possible impact of polypharmacy on HCV treatment is a very interesting topic.
However there are some points to be solved before publication:
Major revision:
1) line 107: the authors used as criterion for staging liver fibrosis a liver stiffness value >12.5 KPa. Did they use Fibroscan or other non invasive techniques for the evaluation of liver fibrosis? Did the authors consider that elevated liver function tests could determine a misclassification?
2) line 109: What do the authors mean? better explain "antiviral therapy was based on physician's criteria". If this sentence does not add more information and criteria used for treatment choice are EASL guidelines, please delete it. If there are additional criteria, please clarify them.
3) It is not clear the need for testing three scales. The aim of the study is to investigate the prevalence of anticholinergic medication and the impact of these drugs on safety.The population included in the "With Ach-drugs" are different by scale.Using different scales with different results are confusing.
How Can the authors draw conclusions? This is very important to use the results in the clinical practice. Can the use of each scale cover specific aspects?
4) Did the authors study a potential association between the use of anticholinergic medication and specific adverse events?
5) The discussion section should be simplified. A shorter section with sentences focused on the aim of the study could improve the readability of the manuscript.
Minor revision:
1) line 60-67: this paragraph should be moved to the discussion section
2) line 83-89: this paragraph should be moved to the discussion section
3) improve the Figure .
4) I suggest that the authors change title the improve readability: "Anticholinergic burden and safety outcomes in older patients with chronic hepatitis C: A retrospective cohort study" could be a possible solution.
Author Response
Report2
Major revision:
1) line 107: the authors used as criterion for staging liver fibrosis a liver stiffness value >12.5 KPa. Did they use Fibroscan or other non invasive techniques for the evaluation of liver fibrosis? Did the authors consider that elevated liver function tests could determine a misclassification?
We thank the reviewer for pointing this out. As stated in the Methods section, cirrhosis was diagnosed based on a liver stiffness value >12.5 kPa (using Fibroscan) in abscence of other markers of portal hypertension (esophageal varices, clinical decompensation or a hepatic venous pressure gradient >= 6 mmHg). Advantages to Fibroscan include a short procedure time (<5 minutes), immediate results, and the ability to perform the test at the bedside or in an outpatient clinic. In patients with chronic HCV infection, regardless the AST/ALT levels, the cut-off of 12.5 kPa has been proposed as the limit to accurately classify cirrhosis with AUROCs between 0.90-0-99 (ref:Castera L et al. GASTROENTEROLOGY 2012;142:1293–1302 ). In addition, all liver stiffness reports were analyzed by the treating physician to corroborate that they accomplished the quality criteria (10 valid measurements, success rate > 60% and IQR/M <= 0.3).
2) line 109: What do the authors mean? better explain "antiviral therapy was based on physician's criteria". If this sentence does not add more information and criteria used for treatment choice are EASL guidelines, please delete it. If there are additional criteria, please clarify them.
As suggested by the reviewer, this sentence has been deleted.
3) It is not clear the need for testing three scales. The aim of the study is to investigate the prevalence of anticholinergic medication and the impact of these drugs on safety. The population included in the "With Ach-drugs" are different by scale. Using different scales with different results are confusing.
How Can the authors draw conclusions? This is very important to use the results in the clinical practice. Can the use of each scale cover specific aspects?
We thank the reviewer for pointing this out. As suggested by Graves-Morris et al. (Front. Pharmacol. 2020;11:570), the use of multiple anticholinergic burden measures on the same population reduces risk from heterogenicity and increase confidence in making comparisons. In order to improve the manuscript, this explanation has been added to both introduction and discussion sections.
As suggested by Hanlon et al. (Ann. Fam. Med. 2020, 18), anticholinergic risk scales are easy and useful to identify patients at risk of adverse effects, regardless of the scale used and Anticholinergic medications are a potentially modifiable risk factor for the prevention of adverse events and one may hypothesize that recognizing the use of anticholinergic drugs, by means of these scales, and therefore potential inappropriate polypharmacy could help in identifying older patients with comorbidities at risk of adverse events when starting antiviral therapy. This information has been added to the discussion section.
4) Did the authors study a potential association between the use of anticholinergic medication and specific adverse events?
As requested by the reviewer we have analyzed the safety profile of patients’ treatment assessing the effects probably attributable to anticholinergics and introduction, methods, results and discussion have been updated including information about anticholinergic effects.
5) The discussion section should be simplified. A shorter section with sentences focused on the aim of the study could improve the readability of the manuscript.
The discussion section has been fully revised according to the reviewer's comments. It has been simplified and relevant information related to safety and anticholinergic events has been added, as recommended by the reviewer.
Minor revision:
1) line 60-67: this paragraph should be moved to the discussion section
Following the reviewer's recommendations, the structure of the paragraph has been changed and removed from the introduction, adding part of the text to the discussion.
2) line 83-89: this paragraph should be moved to the discussion section
According to reviewer's recommendations, this paragraph has been removed from the introduction and moved to the discussion section.
3) improve the Figure .
The figure has been improved to allow better readiness.
4) I suggest that the authors change title the improve readability: "Anticholinergic burden and safety outcomes in older patients with chronic hepatitis C: A retrospective cohort study" could be a possible solution.
Thank you very much for your suggestion. The title has been changed: "Anticholinergic burden and safety outcomes in older patients with chronic hepatitis C: A retrospective cohort study"
Reviewer 3 Report
- Introduction
Please check the sentence at 64-65-66 lines. Can be improved in clarity?
- Methods
- 2.1 Variables
- I Suggest to begin the sentence like this: Patients were….
and omit the first part “In order to reduce bias,”
In my opinion, because this a retrospective study you do not need specify about this kind of bias and probably it is better do not specify at all.
- Results
- 3.1 Baseline characteristics
159,160,161. If I am not wrong, the sum is 135, but the total number of patients is 136. (Maybe there is a reason we can’t understand).
- Please can you clarify the following statement?
Treatment duration was 12 weeks (193, 81.8)
Does this mean that 193 patients had a treatment period of 12 weeks?
Author Response
Report3
- Introduction
Please check the sentence at 64-65-66 lines. Can be improved in clarity?
We appreciate the reviewer pointing this out. The sentence has been removed.
- Methods
2.1 Variables
- I Suggest to begin the sentence like this: Patients were.... and omit the first part “In order to reduce bias,” In my opinion, because this a retrospective study you do not need specify about this kind of bias and probably it is better do not specify at all.
As suggested by the reviewer, the sentence has been reduced to: “Patients had been followed up until 12 weeks after end of treatment with clinical and laboratory evaluations, by the same team of physician, nurse and pharmacist.”
- Results
3.1 Baseline characteristics
159,160,161. If I am not wrong, the sum is 135, but the total number of patients is 136. (Maybe there is a reason we can’t understand).
Thank you for your comment. As the reviewer points out there was a mistake; a patient in the sofosbuvir/simeprevir group was missing.
The sentence has been changed accordingly to “DAA treatment schedule was: 3D±ribavirin (RBV) 118 patients (50.0%), SOF/LDV±RBV 100 patients (42.4%), SOF+RBV 10 patients (4.2%), “SOF/simeprevir (SMV)±RBV 5 patients (2.1%)” and SOF/daclatasvir (DCV)±RBV 3 patients (1.3%).
- Please can you clarify the following statement? Treatment duration was 12 weeks (193, 81.8) Does this mean that 193 patients had a treatment period of 12 weeks?
The treatment duration scheduled was 12 weeks in 193 patients; most of them. Other schemes included 8 weeks (1 patient), 16 weeks (3 patients), 20 weeks (2 patients) and 24 weeks (37 patients).
In order to clarify this point, the sentence has been changed to: “The treatment duration scheduled was 12 weeks in most patients (193 patients, 81.8%).”